# Chemotherapy Response Score in Ovarian Cancer Patients: An Overview of Its Clinical Utility

**DOI:** 10.3390/jcm12062155

**Published:** 2023-03-10

**Authors:** Ioannis Rodolakis, Vasilios Pergialiotis, Michalis Liontos, Dimitrios Haidopoulos, Dimitrios Loutradis, Alexandros Rodolakis, Aristotelis Bamias, Nikolaos Thomakos

**Affiliations:** 1Division of Gynecologic Oncology, 1st Department of Obstetrics and Gynecology, Alexandra Hospital, National and Kapodistrian University of Athens, 10679 Athens, Greece; 2Department of Clinical Therapeutics, Alexandra Hospital, National and Kapodistrian University of Athens, 11527 Athens, Greece

**Keywords:** chemotherapy response, ovarian cancer, interval debulking surgery, progression-free survival, overall survival

## Abstract

The chemotherapy response score has been developed over the last few years as a predictive index of survival outcomes for patients with advanced-stage epithelial ovarian cancer undergoing interval debulking surgery. While its importance in predicting patients at risk of developing recurrences earlier seems to be important, its accuracy in determining patients with a shorter overall survival remains arbitrary. Moreover, standardization of the actual scoring system that was initially developed as a 6-tiered score and adopted as a 3-tiered score is still needed, as several studies suggest that a 2-tiered system is preferable. Given its actual importance in detecting patients with shorter progression-free survival, research should also focus on the actual predictive value of determining patients with platinum resistance, as a suboptimal patient response to standard neoadjuvant chemotherapy might help determine patients at risk of an earlier recurrence. In the present review, we summarize current knowledge retrieved from studies addressing outcomes related to the chemotherapy response score in epithelial ovarian cancer patients undergoing neoadjuvant chemotherapy and discuss differences in outcome reporting to help provide directions for further research.

## 1. Introduction

Ovarian cancer is the leading cause of death among gynecological cancer patients and the fourth most common gynecologic malignancy, accounting for approximately 229,000 new cases on a yearly basis on an international level and more than 150,000 deaths [1]. Its rates are particularly higher in Eastern and Central Europe, with a prevalence that ranges between 6 and 12 cases per 100,000 women, while Asia seems to range lower, with estimated rates of 4 cases per 100,000 women [2]. Primary, complete tumor debulking surgery (PDS) followed by adjuvant chemotherapy is traditionally thought to be the best option for the majority of cases of epithelial ovarian cancer because studies show that it is associated with significantly increased progression-free and overall survival rates [3,4]. However, to ensure optimal survival outcomes, complete tumor cytoreduction is of paramount importance, as suboptimal debulking surgery seems to be accompanied by significant mortality rates that reduce survival to a level that is statistically unsignificant compared to that of patients not offered surgery [5]. The ability of institutions to achieve optimal cytoreduction significantly differs worldwide, with an estimated range between 20% and 90% of cases [6]. The advances in surgical knowledge and expertise permit the completion of operations with a high surgical complexity score in our era [7,8]. When surgery is not considered feasible in terms of ensuring complete tumor resection, current consensus statements for ovarian cancer treatment state that neo-adjuvant chemotherapy should be provided, which must be followed by interval debulking surgery (IDS) [9].

To date, several studies have been published that indicate that the survival rates of patients undergoing IDS are comparable to those of patients undergoing PDS, including three large randomized controlled trials (the SCORPION, the EORTC 55971, and the CHORUS trials) [10,11]. The efficacy of neoadjuvant chemotherapy in reducing tumor load has been addressed in several trials, and the results seem to be promising for the majority of patients, as perioperative morbidity in the IDS setting seems to have decreased [12]. Tumor reduction following chemotherapy seems to differentiate among cases of epithelial ovarian cancer. It is observed that even patients assigned to the most common high-grade serous histology seem to differ considerably in terms of achieving a partial or even complete tumor response. Histopathologically, the tumor response to neoadjuvant chemotherapy (NACT) presents with sites of tumor fibrosis, macrophage infiltration, tumor-induced inflammation, and eventually necrosis. The chemotherapy response score (CRS) has been instituted as a pathology index that might help determine prognosis and guide decision-making in epithelial ovarian cancer [13].

A previous meta-analysis in this field indicated that patients with a complete chemotherapy response had significantly better survival rates compared to those with a partial or no response [14]. However, significant heterogeneity was observed in the patient population as well as in the methodology that was used to assess the CRS score. In the present communication paper, we gathered the available information to comment on the potential gaps that arise from the methodological differences of published studies and performed a systematic review of the literature to identify new articles.

## 2. Rationale for the Development of the CRS

Until the development of the CRS, patients’ responses to neoadjuvant chemotherapy were evaluated mainly with macroscopic criteria, namely, differences in the extent of dissemination and maximal tumor size. The biochemical response is also tracked using serum tumor markers, including cancer antigen 125 and human epididymis 4. However, several questions arose that could not be answered, including which tissue should be considered as the main feature that could help distinguish patients with partial and complete responses (ovaries, omentum, or other extraovarian sites, including peritoneal surfaces) and which feature could help prognostically distinguish between patients with a favorable and unfavorable outcome (extent of regression, pattern of residual disease, etc.). 

The development of the CRS aimed to provide an accurate scoring system that would permit an objective pathologist’s evaluation of the actual tumor response to chemotherapy. It is considered to be very precise, as the reported interobserver reproducibility seems to be exceptionally high, according to the findings of a previous study [15]. It is based on a 3-tiered system that evaluates the proportion of viable tumor cells and the presence of regression-associated fibroinflammatory changes. Therefore, each patient is assigned a score of CRS1, CRS2, or CRS3 that corresponds to no/minimal, partial, or complete response, respectively.

## 3. Review Methodology

### Design and Eligibility Criteria

To detect new studies published after the publication of the previous systematic review [16], we systematically searched the literature while considering the current recommendations of the Preferred Reporting Items for Systematic Reviews and Meta-Analyses (PRISMA) guidelines [17]. Eligibility criteria for the inclusion of studies were predetermined. We chose to include observational studies and randomized trials that compared differences in survival among patients undergoing IDS for whom evaluation of the CRS was performed based on the pathology analysis. The review was designed to include all studies that were published in the Medline (1966–2023), Scopus (2004–2023), Clinicaltrials.gov (2008–2023), EMBASE (1980–2023), and Cochrane Central Register of Controlled Trials CENTRAL (1999–2023) databases. The date of the last search was set to 31 January 2023. Six articles were finally retrieved [16,18,19,20,21,22], and their methodology is discussed in the present paper.

## 4. Chemotherapy Response Score, Survival Outcomes, and Platinum Resistance

Bohm et al. were the first to evaluate the prognostic accuracy of CRS on the PFS of epithelial ovarian cancer patients [18]. They recruited 133 patients, of whom 62 represented the test cohort and 71 the validation cohort. In this study, researchers evaluated the chemotherapy response based on an earlier 6-tiered system and noticed significant differences among best responders (CRS4 and CRS5) and intermediate responders (CRS2 and CRS3) (median survival, 32.1 vs. 11.3 months, respectively). They also observed that by condensing the score to three tiers, the reproducibility increased considerably. Furthermore, the authors reported that reduction of serum CA-125 levels following neoadjuvant chemotherapy was not predictive of the patients’ CRS score. This implies that the latter should be independently evaluated and may substitute for serum markers in terms of defining patients at risk of early relapse.

Ditzel et al. confirmed these results, denoting that patients with a CRS1 or CRS2 (on a 3-tiered system) had considerably shorter progression-free survival compared to patients with a CRS3 (18.9 vs. 10.9 months) [19]. The authors noted that the reproducibility of findings was the best when the response of omental metastases was considered, whereas the agreement of pathologists regarding the CRS of adnexal masses was considerably lower, indicating that reporting should be based on omental histology slides. 

Lawson et al. further decreased the subgrouping of patients, supporting the use of a binary system rather than a 3-tiered score, as they observed that the omental CRS was significantly associated with PFS only when they divided patients into incomplete/partial and complete responses [20]. They also reported that both the 2-tiered and 3-tiered systems of adnexal scoring were associated with the PFS; however, they did not investigate the reproducibility of findings. 

Another study evaluated 115 stage IIIC and stage IV epithelial ovarian cancer patients and reported comparable reproducibility among the omental and the adnexal CRSs [21]. In their series, they noted significant differences in PFS among patients with a CRS1 and CRS2 as well as among patients with a CRS2 and CRS3. However, the total number of chemotherapy cycles that were used was particularly heterogeneous, as only 33.6% of patients received the recommended total of 6 doses, whereas nearly 61% of patients received >6 doses. They were also the first to notice that differences in the percentage of patients achieving optimal cytoreduction (residual tumor < 1 cm) were not significant. They did notice, however, a trend towards increased platinum resistance among CRS1-2 (31.8%) and CRS3 (23.4%) patients, although the result was not significant (*p =* 0.336). 

Michaan et al. confirmed the association of the CRS’ 3-tiered grading in omental and ovarian biopsies with the PFS; however, no correlation was observed with the OS [22]. Agreements in ovarian tissue (85%) and omental (97.5%) biopsy scoring were observed among pathologists. 

Lastly, Santoro et al. observed that both CRS scores 1 and 2 in the omental and ovarian biopsies were associated with the PFS and OS rates [23]. The investigation of scoring values in other extraovarian sites also revealed significant differences between cases with and without residual diseases.

## 5. Discussion

Current evidence published in the international literature suggests that the CRS may be considered an important predictive factor of progression-free survival in patients who receive neoadjuvant chemotherapy for advanced ovarian cancer at the initial stage. However, it remains arbitrary whether the score can predict patients with improved overall survival or the actual possibility of platinum resistance. 

Moreover, it should be highlighted that scoring of ovarian or extraovarian tissues other than the omentum is not accompanied by a reproducibility that seems to be acceptable for current clinical practice; hence, until further evidence becomes available, the CRS should be evaluated only from pathology reviews involving the omentum. 

To date, at least two meta-analyses have been published that are related to the predictive value of the CRS on the survival rates of patients with epithelial ovarian cancer [14,16]. While both of them suggest that a CRS3 is associated with superior PFS rates compared to a CRS1-2, it should be stressed out that both of them are based on studies that include patients with both optimal and suboptimal tumor cytoreduction, as well as patients that received a standard 6-cycle chemotherapy regimen and patients that received >6 cycles. Given that these variables may considerably affect survival rates and the fact that none of the included studies had propensity score matching that could partially help diminish selection bias, it should be emphasized that there is a possibility of skewness in the actual findings of published individual studies and systematic reviews.

### Implications for Current Clinical Practice and Future Research

Given the actual data provided by the published studies, we believe that the CRS of omental specimens should be viewed as potentially predictive of the progression-free and overall survival rates of epithelial ovarian cancer patients. This information may guide decision-making concerning the actual intervals of follow-up visits, although a consensus is still lacking. 

We believe that further research is needed concerning the index’s accuracy in predicting which patients may develop platinum resistance. This information will help with decision-making about the actual chemotherapies that epithelial ovarian cancer patients with CRS1 need to take. Furthermore, a binary system has been shown to be more informative and applicable in clinical settings in comparison to the widely reported 3-tiered system. Since epithelial ovarian cancer cells gradually become polyclonal and the genetic heterogeneity of metastatic lesions has already been established [24,25], it would be prudent to evaluate the actual summative CRS in various sites, including the omentum, ovarian tissue, and peritoneal disease. This way, future research might be able to identify patients with a favorable CRS score from omental biopsies who actually have less responsive extra-omental lesions and, therefore, a higher tendency to relapse. 

## 6. Conclusions

Evaluation of the chemotherapy response score of omental metastatic lesions seems to be associated with survival outcomes in epithelial ovarian cancer patients. However, it remains unclear to this date if this information could guide decision-making in clinical practices, as the methodological heterogeneity and presented results of published studies provide firm conclusions concerning its impact on survival outcomes. We believe that future research will help increase our insight on the actual clinical importance of this histopathological index as further associations become available and specific groups of patients are analyzed accordingly. 

## Data Availability

The data are available upon request from the corresponding author.

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
