# Peer review of "Chemotherapy Response Score in Ovarian Cancer Patients: An Overview of Its Clinical Utility"

_jcm, 2023, doi:10.3390/jcm12062155_

Round 1
Reviewer 1 Report
The communication of Rodolakis makes an active contribution to the discussion on the possibility to use the evaluation chemotherapy response as a predictive index of survival outcomes. For this reason, it may be of interest for the vast audience of clinicians that deal with ovarian cancer
Only, I recommend the authors to check the sentence reported at the top of page 2 “ To date, several studies have been published that indicate that survival rates of patients undergoing IDS are comparable to those of patients undergoing IDS” ) which is surely incorrect.
Author Response
Dear reviewer,
we thank you for reviewing our communication letter and for your suggestion. Below you will find answered your comments.
The communication of Rodolakis makes an active contribution to the discussion on the possibility to use the evaluation chemotherapy response as a predictive index of survival outcomes. For this reason, it may be of interest for the vast audience of clinicians that deal with ovarian cancer.
Only, I recommend the authors to check the sentence reported at the top of page 2 “ To date, several studies have been published that indicate that survival rates of patients undergoing IDS are comparable to those of patients undergoing IDS” ) which is surely incorrect.
Authors reply: we thank the reviewer for this remark. The phrase has been revised accordingly (Line 76).
Reviewer 2 Report
Dear all, I think it is a very hot topic nowadays in ovarian cacner. To boost the soundness of the article I 'd advice to do a systematic review of the topic, it should not be very difficult because there are not so many publications on CRS in ovarian cacner , as far as I know. KR,
Author Response
Dear reviewer,
we thank you for reviewing our communication letter and for your suggestion. Below you will find answered your comments.
Reviewer 2
Dear all, I think it is a very hot topic nowadays in ovarian cacner. To boost the soundness of the article I 'd advice to do a systematic review of the topic, it should not be very difficult because there are not so many publications on CRS in ovarian cancer , as far as I know.
Authors reply: we thank the reviewer for this remark. This study is submitted as a communication letter with the primary aim to discuss the methodological heterogeneity of published studies that significantly limits a synthetic review (meta-analysis). Previous meta-analyses have been already published in the field and the present communication is based in a systematic review of the literature with the aim to detect all published research. However, given the limited amount of evidence, the existence of previous meta-analyses that have already included the studies discussed in this article and the absence of novel evidence we opted to focus only in the methodological heterogeneity and provide directions for future research. (Lines 89-96 and 118-129)
Reviewer 3 Report
In the present study, the authors focused on the chemotherapy response score (CRS) in epithelial ovarian cancer patients undergoing neoadjuvant chemotherapy and interval debulking surgery. They mainly listed six studies that discuss number of tiers, reported sites, and their correlation with prognosis indexes. Altogether, this paper provides an overview of the effects of CRS in advanced stage epithelial ovarian cancer. However, This review paper is a simplified review with limited information, and it does not provide new information and perspectives.
Major points:
1. Regarding the writing, some sentences are too long to understand the actual meaning easily.
2. Paragraph one, page two. “To date, several studies have been published that indicate that survival rates of patients undergoing IDS are comparable to those of patients undergoing IDS.” This statement is confusing.
Author Response
Dear reviewer,
we thank you for reviewing our communication letter and for your comments. Below you will find our answers.
Sincerely,
the authors.
In the present study, the authors focused on the chemotherapy response score (CRS) in epithelial ovarian cancer patients undergoing neoadjuvant chemotherapy and interval debulking surgery. They mainly listed six studies that discuss number of tiers, reported sites, and their correlation with prognosis indexes. Altogether, this paper provides an overview of the effects of CRS in advanced stage epithelial ovarian cancer. However, This review paper is a simplified review with limited information, and it does not provide new information and perspectives.
Authors reply: we thank the reviewer for this remark. This is a communication letter that aims to discuss the significant methodological differences of published studies. While we acknowledge the fact that all the data have been previously presented by researchers we believe that a special focus on the differences will help future research provide direct answers to questions that still remain in this field.
Major points:
- Regarding the writing, some sentences are too long to understand the actual meaning easily.
Authors reply: several phrases were revised in the present revision. If further revision is deemed necessary we will be happy to revise accordingly.
- Paragraph one, page two. “To date, several studies have been published that indicate that survival rates of patients undergoing IDS are comparable to those of patients undergoing IDS.” This statement is confusing.
Authors reply: we thank the reviewer for this remark. The phrase has been revised accordingly (Line 76).
Round 2
Reviewer 3 Report
Thank you for the authors‘ response, and my questions have been addressed.
As a communication paper, I considered the current manuscript has been greatly improved.